# Hierarchical Graph Transformer with Adaptive Node Sampling

**Zaixi Zhang**[1,2] **Qi Liu**[1,2,*] **Qingyong Hu**[3], **Chee-Kong Lee**[4]
1: Anhui Province Key Lab of Big Data Analysis and Application,
School of Computer Science and Technology, University of Science and Technology of China
2:State Key Laboratory of Cognitive Intelligence, Hefei, Anhui, China
3:Hong Kong University of Science and Technology, 4: Tencent America
zaixi@mail.ustc.edu.cn, qiliuql@ustc.edu.cn
qhuag@cse.ust.hk, cheekonglee@tencent.com

## Abstract

The Transformer architecture has achieved remarkable success in a number of domains including natural language processing and computer vision. However, when it comes to graph-structured data, transformers have not achieved competitive performance, especially on large graphs. In this paper, we identify the main deficiencies of current graph transformers: (1) Existing node sampling strategies in Graph Transformers are agnostic to the graph characteristics and the training process. (2) Most sampling strategies only focus on local neighbors and neglect the long-range dependencies in the graph. We conduct experimental investigations on synthetic datasets to show that existing sampling strategies are sub-optimal. To tackle the aforementioned problems, we formulate the optimization strategies of node sampling in Graph Transformer as an adversary bandit problem, where the rewards are related to the attention weights and can vary in the training procedure. Meanwhile, we propose a hierarchical attention scheme with graph coarsening to capture the long-range interactions while reducing computational complexity. Finally, we conduct extensive experiments on real-world datasets to demonstrate the superiority of our method over existing graph transformers and popular GNNs.

## 1 Introduction

In recent years, the Transformer architecture [33] and its variants (e.g., Bert [7] and ViT [8]) have achieved unprecedented successes in natural language processing (NLP) and computer vision (CV). In light of the superior performance of Transformer, some recent works [21, 38] attempt to generalize Transformer for graph data by treating each node as a token and designing dedicated positional encoding. However, most of these works only focus on small graphs such as molecular graphs with tens of atoms [38]. For instance, Graphormer [38] achieves state-of-the-art performance on molecular property prediction tasks. When it comes to large graphs, the quadratic computational and storage complexity of the vanilla Transformer with the number of nodes inhibits the practical application. Although some Sparse Transformer methods [30, 2, 19] can improve the efficiency of the vanilla Transformer, they have not exploited the unique characteristics of graph data and require a quadratic or at least sub-quadratic space complexity, which is still unaffordable in most practical cases. Moreover, the full-attention mechanism potentially introduces noise from numerous irrelevant nodes in the full graph.

To generalize Transformer to large graphs, existing Transformer-based methods [9, 46, 7] on graphs explicitly or implicitly restrict each node's receptive field to reduce the computational and storage

---

*Qi Liu is the corresponding author.

36th Conference on Neural Information Processing Systems (NeurIPS 2022).

complexity. For example, Graph-Bert [41] restricts the receptive field of each node to the nodes with top-$k$ intimacy scores such as Personalized PageRank (PPR). GT-Sparse [9] only considers 1-hop neighboring nodes. We argue that existing Graph Transformers have the following deficiencies: (1) The fixed node sampling strategies in existing Graph Transformers are ignorant of the graph properties, which may sample uninformative nodes for attention. Therefore, an adaptive node sampling strategy aware of the graph properties is needed. We conduct case studies in Section 4 to support our arguments. (2) Though the sampling method enables scalability, most node sampling strategies focus on local neighbors and neglect the long-range dependencies and global contexts of graphs. Hence, incorporating complementary global information is necessary for Graph Transformer.

To solve the challenge (1), we propose **A**daptive **N**ode **S**ampling for **G**raph **T**ransformer (ANS-GT) and formulate the optimization strategy of node sampling in Graph Transformer as an adversary bandit problem. Specifically in ANS-GT, we modify Exp4.P method [3] to adaptively assign weights to several chosen sampling heuristics (e.g., 1-hop neighbors, 2-hop neighbors, PPR) and combine these sampling strategies to sample informative nodes. The reward is proportional to the attention weights and the sampling probabilities of nodes, i.e. the reward to a certain sampling heuristic is higher if the sampling probability distribution and the node attention weights distribution are more similar. Then in the training process of Graph Transformer, the node sampling strategy is updated simultaneously to sample more informative nodes. With more informative nodes input into the self-attention module, ANS-GT can achieve better performance.

To tackle the challenge (2), we propose a hierarchical attention scheme for Graph Transformer to encode both local and global information for each node. The hierarchical attention scheme consists of fine-grained local attention and coarse-grained global attention. In the local attention, we use the aforementioned adaptive node sampling strategy to select informative local nodes for attention. As for global attention, we first use graph coarsening algorithms [26] to pre-process the input graph and generate a coarse graph. Such algorithms mimic a down-sampling of the original graph via grouping the nodes into super-nodes while preserving global graph information as much as possible. The center nodes then interact with the sampled super-nodes. Such coarse-grained global attention helps each node capture long-distance dependencies while reducing the computational complexity of the vanilla Graph Transformers.

We conduct extensive experiments on real-world datasets to show the effectiveness of ANS-GT. Our method outperforms all the existing Graph Transformer architectures and obtains state-of-the-art results on 6 benchmark datasets. Detailed analysis and ablation studies further show the superiority of the adaptive node sampling module and the hierarchical attention scheme.

In summary, we make the following contributions:

- We propose **A**daptive **N**ode **S**ampling for **G**raph **T**ransformer (ANS-GT), which modifies a multi-armed bandit algorithm to adaptively sample nodes for attention.

- In the hierarchical attention scheme, we introduce coarse-grained global attention with graph coarsening, which helps graph transformer capture long-range dependencies while increasing efficiency.

- We empirically evaluate our method on six benchmark datasets to show the advantage over existing Graph Transformers and popular GNNs.

## 2 Related Work

### 2.1 Transformers for Graph

Recently, Transformer [33] has shown its superiority in an increasing number of domains [7, 8, 40], e.g. Bert [7] in NLP and ViT [8] in CV. Existing works attempting to generalize Transformer to graph data mainly focus on two problems: (1) How to design dedicated positional encoding for the nodes; (2) How to alleviate the quadratic computational complexity of the vanilla Transformer and scale the Graph Transformer to large graphs. As for the positional encoding, GT [9] firstly uses Laplacian eigenvectors to enhance node features. Graph-Bert [41] studies employing Weisfeiler-Lehman code to encode structural information. Graphormer [38] utilizes centrality encoding to enhance node features while incorporating edge information with spatial (SPD-indexed attention bias) and edge encoding. SAN [21] further replaces the static Laplacian eigenvectors with learnable positional encodings and

designs an attention mechanism that distinguishes local connectivity. For the scalability issue, one immediate idea is to restrict the number of attending nodes. For example, GAT [34] and GT-Sparse [9] only consider the 1-hop neighboring nodes; Gophormer [46] uses GraphSAGE [11] sampling to uniformly sample ego-graphs with pre-defined maximum depth; Graph-Bert [41] restricts the receptive field of each node to the nodes with top-k intimacy scores (e.g., Katz and PPR). However, these fixed node sampling strategies sacrifice the advantage of the Transformer architecture. SAC [22] tries to use an LSTM edge predictor to predict edges for self-attention operations. However, the fact that LSTM can hardly be parallelized reduces the computational efficiency of the Transformer.

## 2.2 Sparse Transformers

In parallel, many efforts have been devoted to reducing the computational complexity of the Transformer in the field of NLP [23] and CV [32]. In the domain of NLP, Longformer [2] applies block-wise or strode patterns while only fixing on fixed neighbors. Reformer [19] replaces dot-product attention by using approximate attention computation based on locality-sensitive hashing. Routing Transformer [30] employs online k-means clustering on the tokens. Linformer [35] demonstrates that the self-attention mechanism can be approximated by a low-rank matrix and reduces the complexity from $\mathcal{O}(n^2)$ to $\mathcal{O}(n)$. As for vision transformers, Swin Transformer [24] proposes the shifted windowing scheme which brings greater efficiency by limiting self-attention computation to non-overlapping local windows while also allowing for cross-window connection. Focal Transformer [37] presents a new mechanism incorporating both fine-grained local and coarse-grained global attention to capture short- and long-range visual dependencies efficiently. However, these sparse transformers do not take the unique graph properties into consideration.

## 2.3 Graph Neural Networks and Node Sampling

Graph neural networks (GNNs) [18, 11, 12, 44, 43, 31, 45, 42] follow a message-passing schema that iteratively updates the representation of a node by aggregating representations from neighboring nodes. When generalizing to large graphs, Graph Neural Networks face a similar scalability issue. This is mainly due to the uncontrollable neighborhood expansion in the aggregation stage of GNN. Several node sampling algorithms have been proposed to limit the neighborhood expansion, which mainly falls into node-wise sampling methods and layer-wise sampling methods. In node-wise sampling, each node samples $k$ neighbors from its sampling distribution, then the total number of nodes in the $l$-th layer becomes $\mathcal{O}(k^l)$. GraphSage [11] is one of the most well-known node-wise sampling methods with the uniform sampling distribution. GCN-BS [25] introduces a variance reduced sampler based on multi-armed bandits. To alleviate the exponential neighbor expansion $\mathcal{O}(k^l)$ of the node-wise samplers, layer-wise samplers define the sampling distribution as a probability of sampling nodes given a set of nodes in the upper layer [4, 16, 49]. From another perspective, these sampling methods can also be categorized into fixed sampling strategies [11, 4, 49] and adaptive strategies [25, 39]. However, none of the above sampling methods in GNNs can be directly applied in Graph Transformer as Graph Transformer does not follow the message passing schema.

# 3 Preliminaries

## 3.1 Problem Definition

Let $G = (A, X)$ denote the unweighted graph where $A \in \mathbb{R}^{n \times n}$ represents the symmetric adjacency matrix with $n$ nodes, and $X \in \mathbb{R}^{n \times p}$ is the attribute matrix of $p$ attributes per node. The element $A_{ij}$ in the adjacency matrix equals to 1 if there exists an edge between node $v_i$ and node $v_j$, otherwise $A_{ij} = 0$. The label of node $v_i$ is $y_i$. In the node classification problem, the classifier has the knowledge of the labels of a subset of nodes $V_L$. The goal of semi-supervised node classification is to infer the labels of nodes in $V \backslash V_L$ by learning a classification function.

## 3.2 Transformer Architecture

The Transformer architecture consists of a series of Transformer layers [33]. Each Transformer layer has two parts: a multi-head self-attention (MHA) module and a position-wise feed-forward network (FFN). Let $\mathbf{H} = [\boldsymbol{h}_1, \cdots, \boldsymbol{h}_m]^\top \in \mathbb{R}^{m \times d}$ denote the input to the self-attention module where $d$ is

the hidden dimension, $\boldsymbol{h}_i \in \mathbb{R}^{d \times 1}$ is the hidden representation at position $i$, and $m$ is the number of positions. The MHA module firstly projects the input $\mathbf{H}$ to query-, key-, value-spaces, denoted as $\mathbf{Q}, \mathbf{K}, \mathbf{V}$, using three matrices $\mathbf{W}_Q \in \mathbb{R}^{d \times d_K}, \mathbf{W}_K \in \mathbb{R}^{d \times d_K}$ and $\mathbf{W}_V \in \mathbb{R}^{d \times d_V}$:

$$\mathbf{Q} = \mathbf{H}\mathbf{W}_Q, \quad \mathbf{K} = \mathbf{H}\mathbf{W}_K, \quad \mathbf{V} = \mathbf{H}\mathbf{W}_V. \tag{1}$$

Then, in each head $i \in \{1, 2, \ldots, B\}$ ($B$ is the total number of heads), the scaled dot-product attention mechanism is applied to the corresponding $\{\mathbf{Q}_i, \mathbf{K}_i, \mathbf{V}_i\}$:

$$\text{head}_i = \text{Softmax}\left(\frac{\mathbf{Q}_i \mathbf{K}_i^T}{\sqrt{d_K}}\right) \mathbf{V}_i. \tag{2}$$

Finally, the outputs from different heads are further concatenated and transformed to obtain the final output of MHA:

$$\text{MHA}(\mathbf{H}) = \text{Concat}\left(\text{head}_1, \ldots, \text{head}_B\right) \mathbf{W}_O, \tag{3}$$

where $\mathbf{W}_O \in \mathbb{R}^{d \times d}$. In this work, we employ $d_K = d_V = d/B$.

### 3.3 Graph Coarsening

The goal of Graph Coarsening [29, 26, 17] is to reduce the number of nodes in a graph by clustering them into super-nodes while preserving the global information of the graph as much as possible. Given a graph $G = (V, E)$ ($V$ is the node set and $E$ is the edge set), the coarse graph is a smaller weighted graph $G' = (V', E')$. $G'$ is obtained from the original graph by first computing a partition $\{C_1, C_2, \cdots, C_{|V'|}\}$ of $V$, i.e., the clusters $C_1 \cdots C_{|V'|}$ are disjoint and cover all the nodes in $V$. Each cluster $C_i$ corresponds to a super-node in $G'$ The partition can also be characterized by a matrix $\hat{P} \in \{0, 1\}^{|V| \times |V'|}$, with $\hat{P}_{ij} = 1$ if and only if node $v_i$ in $G$ belongs to cluster $C_j$. Its normalized version can be defined by $P \triangleq \hat{P}D^{-\frac{1}{2}}$, where $D$ is a $|V'| \times |V'|$ diagonal matrix with $|C_i|$ as its $i$-th diagonal entry. The feature matrix and weighted adjacency matrix of $G'$ are defined by $X' \triangleq P^T X$ and $A' \triangleq P^T A P$. After Graph Coarsening, the number of nodes/edges in $G'$ is significantly smaller than that of $G$. The coarsening rate can be defined as $c = \frac{|V'|}{|V|}$.

## 4 Motivating Observations

To generalize Transformer to large graphs, existing Graph Transformer models typically choose to sample a batch of nodes for attention. However, real-world graph datasets exhibit different properties, which makes a fixed node sampling strategy unsuitable for all kinds of graphs. Here, we present a simple yet intuitive case study to illustrate how the performance of Graph Transformer changes with different node sampling strategies. The main idea is to use four popular node sampling strategies for node classification: 1-hop neighbors, 2-hop neighbors, PPR, and KNN. Then, we will check the performance of Graph Transformer on graphs with different properties. If the performance drops sharply when the property varies, it will indicate that graphs with different properties may require different node sampling strategies.

We conduct experiments on the Newman artificial networks [10] since it enable us to obtain networks with different properties easily. More detailed settings can be found in Appendix A. Here, we consider the property of homophily/heterophily as one example. Following previous works [47], the degree of homophily $\alpha$ can be defined as the fraction of edges in a network connecting nodes with the same class label, i.e. $\alpha \triangleq \frac{|(v_i, v_j) \in E \wedge y_i = y_j|}{|E|}$. Graphs with $\alpha$ closer to 1 tend to have more edges connecting nodes within the same class, i.e. strong homophily; whereas networks with $\alpha$ closer to 0 have more edges connecting nodes in different classes, i.e. strong heterophily.

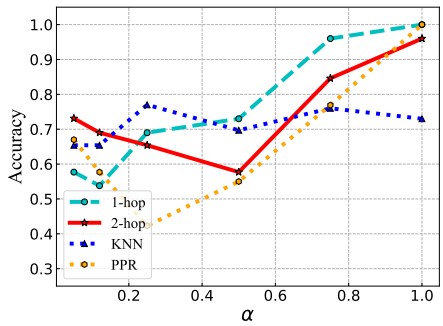

Figure 1: Performance of Graph Transformer using different node sampling mechanisms: 1-hop, 2-hop, PPR and kNN respectively on Newman networks.

As shown in Figure 1, there is no sampling strategy that dominates other strategies across the spectrum of

$\alpha$, which supports our claim that an adaptive sampling
method is needed for graphs with different properties. For graphs with strong homophily (e.g., $\alpha = 1.0$), it is easy to obtain high accuracy by sampling 1-hop neighbors or nodes with top PPR scores. On the other hand, for graphs with strong heterophily (e.g., $\alpha = 0.05$), the accuracy of using 2-hop neighbors as neighborhoods (i.e., 73.1%) is much higher than that of using 1-hop neighbors and PPR. The probable reason is that the homophily ratio of 2-hop neighbors may rise with the increase of inter-class edges. Finally, Graph Transformer with KNN node sampling can get the most consistent results since KNN only calculates the similarity of attributes. Thus, KNN achieves the best performance (i.e., 77.2% accuracy) when all the nodes are connected randomly, i.e. $\alpha = 0.25$ for the Newman network.

Hence, considering different graph properties (e.g., homophily/heterophily), Graph Transformer should adaptively sample the most informative nodes for attention. These observations motivate us to design the proposed hierarchical Graph Transformer with adaptive node sampling in Section 5.

## 5  The Proposed Method

In light of the limitations of existing Graph Transformers for large graphs and the motivating observation in the previous section, we propose two effective methods for Graph Transformer to adaptively sample informative nodes and capture the long-range coarse-grained dependencies in this section. We also show that our method has a computational complexity of $\mathcal{O}(n)$. The overview of the model framework is shown in Figure 2.

### 5.1  Adaptive Node Sampling

Our Adaptive Node Sampling module aims to adaptively choose the batch of most informative nodes by a multi-armed bandit mechanism. In our setting, it is intuitive that the contributions of nodes to the learning performance can be time-sensitive. Meanwhile, the rewards which are associated with the model training process are not independent random variables across the iterations. The above situation can satisfy the adversarial setting in the multi-armed bandit problem [1]. To adaptively choose the most informative nodes with the designed sampling strategies, we adjust the method ALBL proposed in [14], which modifies the EXP4.P method [3]. EXP4.P possesses a strong theoretical guarantee for the adversarial setting.

Formally, let $w^t = (w_1^t, \cdots, w_K^t)$ be the adaptive weight vector in iteration $t$, where the $k$-th non-negative element $w_k^t$ is the weight corresponding to the $k$-th node sampling strategy. The weight vector $w^t$ is then scaled to a probability vector $p^t = (p_1^t, \cdots, p_K^t)$ where $p_k^t \in [p_{min}, 1]$ with $p_{min} > 0$. Our ANS-GT adaptively sample nodes based on the probability vector and then obtains the reward of the action.

For each center node, we consider the sampling probability matrix $Q^t \in \mathbb{R}^{K \times n}$, where $K$ is the number of sampling heuristics and $n$ is the number of nodes in the graph. Specifically, $Q_{k,j}^t$ denotes the $k$-th sampling strategy's preference on selecting node $j$ in iteration $t$ and $Q^t$ is normalized to satisfy $\sum_{j=1}^n Q_{k,j}^t = 1$. Note that our ANS-GT is a general framework and is not restricted to a certain set of sampling heuristics. In our work, we adopt four representative sampling heuristics:

**1-/2-hop neighbors**: We adopt the normalized adjacency matrix $\widetilde{A} = \hat{D}^{-\frac{1}{2}} \hat{A} \hat{D}^{-\frac{1}{2}}$ for 1-hop neighbors and $\widetilde{A}^2$ for 2-hop neighbors, where $\hat{A} = A + I$ is the adjacency matrix of the graph $G$ with self connections added and $\hat{D}$ is a diagonal matrix with $\hat{D}_{ii} = \sum_j \hat{A}_{ij}$.

**KNN**: We adopts the cosine similarity of node attributes to measure the similarities of nodes. Mathematically, the similarity score $S_{ij}$ between node $i$ and $j$ is calculated as $S_{ij} = x_i \cdot x_j / (|x_i| \cdot |x_j|)$ where $x_i$ is the feature vector of node $v_i$.

**PPR**: The Personalized PageRank [27] matrix $S$ is calculated as: $S = c(I - (1 - c)\overline{A})^{-1}$, where factor $c \in [0, 1]$ (set to 0.15 in our experiments). $\overline{A}$ denotes the column-normalized adjacency matrix.

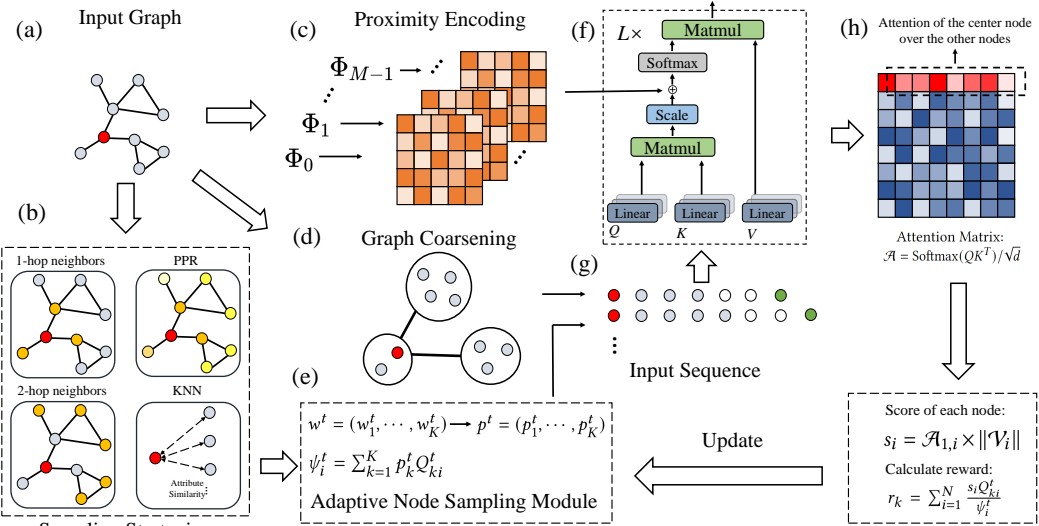

Figure 2: Model framework of our proposed method: (a) An example input graph. The center node for sampling is colored red. (b) We consider four sampling strategies in this work, i.e. 1-hop neighbors, 2-hop neighbors, PPR, and KNN. (c) The proximity encoding module. (d) Graph coarsening to cluster nodes into super-nodes. (e) The adaptive node sampling module. (f) The self-attention module in Graph Transformer. The output node embeddings are used for node classification. (g) In the sampled input node sequences, the gray nodes are the fine-grained nodes; the white nodes are the coarse-grained nodes from graph coarsening; the green nodes denote the global nodes. (h) We use the first row of the attention matrix, i.e., $\mathcal{A}_{1,i}$ multiplying the magnitude of the corresponding value $V_i$ to represent the significance of each node. Then we calculate the reward for each sampling strategy and update the weights.

Finally, given the probability vector $p^t$ and the node sampling matrices $Q^t$, the final node sampling probability is:

$$\psi_i^t = \sum_{k=1}^{K} p_k^t Q_{ki}^t. \tag{4}$$

We introduce a novel reward scheme based on the attention weights which is intrinsic in Transformer. Formally, given an attention matrix $\mathcal{A} = \text{Softmax}(QK^T)/\sqrt{d}$, we use the first row of the attention matrix, i.e., $\mathcal{A}_{1,i}$ multiplying the magnitude of corresponding value $V_i$ to represent the significance of each node to the center node: $s_i = \mathcal{A}_{1,i} \times \|\mathcal{V}_i\|$. In the situation of multiple heads and layers in the Transformer, we average the significance scores in multiple attention matrices. The reward to the $k$-th sampling strategy is: $r_k = \sum_{i=1}^{N} \frac{s_i Q_{ki}^t}{\psi_i^t}$, where $N$ is the number of sampled nodes for each center node. $r_k$ can be interpreted as the dot product between the significance score vector and the normalized sampling probability vector. The intuition behind the reward design is that the reward to a certain sampling heuristic is higher if the sampling probability distribution and the node significance score distribution are closer. Thus, exploiting such a sampling heuristic can help graph transformer sample more informative nodes. Finally, we update $w^t$ with the reward. In experiments, for the efficiency and stability of training, we update the sampling weights and resample nodes every $T$ epochs. The pseudo-code of ANS-GT is listed in Algorithm 1.

## 5.2 Hierarchical Graph Attention

We argue that most node sampling strategies (e.g., 1-hop neighbors) focus on local information and neglect long-range dependencies or global contexts. Therefore, to efficiently capture both the local and global information in the graph, we propose a novel hierarchical graph attention scheme including fine-grained attention and coarse-grained attention. Specifically, we use the proposed adaptive node sampling for local fined-grained attention. On the other hand, we adopt the graph

---

**Algorithm 1** ANS-GT

---

**Input**: Total training epochs $E$; $p_{min}$; update period $T$; the number of sampled nodes $N$.
**Output**: Trained Graph Transformer model, optimized $w^t$.
  1: Set $w_k^1 = 1$ for $k = 1, \cdots, K$.
  2: Calculate the sampling probability matrix $Q^t$.
  3: **for** $t = 1, 2, \cdots, E$ **do**
  4:    Train Graph Transformer with the sampled node sequences.
  5:    **if** $t\%T = 0$. **then**
  6:       Obtains the attention matrices and calculate the significance scores: $s_i = \mathcal{A}_{1,i} \times \|\mathcal{V}_i\|$.
  7:       Set $W^t = \sum_{k=1}^{K} w_k^t$, and set $p_k^t = (1 - Kp_{min})\sum_{j=1}^{K} \frac{w_j^t}{W^t} + p_{min}$ for $k = 1, \cdots, K$.
  8:       Calculate $\psi_i^t$ in Equ. 4 and sample $N$ nodes.
  9:       Set $r_k = \sum_{i=1}^{N} \frac{s_i Q_{ki}^t}{\psi_i^t}$ and update the weight vector $w_k^{t+1}$ using
$$w_k^{t+1} = w_k^t e^{(\frac{p_{min}}{2})(r_k + \frac{1}{P_k^t})\sqrt{\frac{ln(N/0.1)}{KT}}}.$$
 10:    **end if**
 11: **end for**

---

coarsening algorithm [26] to generate the coarsened graph $G'$. The sampled $n_s$ super-nodes from $G'$ are used to capture long-range dependencies. Similar to [46], we also use $n_g$ global nodes with learnable features to store global context information. Finally, the hierarchical nodes are concatenated with the center nodes as the input sequences.

For the positional encoding, we use the proximity encoding in [46]: $\Phi_m(v_i, v_j) = \widetilde{A}^m[i, j], m \in \{0, \cdots, M - 1\}$, where $\widetilde{A}$ denotes the normalized adjacency matrices with self-loop. Note that our framework is agnostic to the positional encoding scheme and we left other positional encoding methods such as Laplacian eigenvectors [9] for future exploration. We follow the Graphormer framework to obtain the output of the $l$-th transformer layer, $\mathbf{H}^{(l)}$:

$$\hat{\mathbf{H}}^{(l-1)} = \text{MHA}(\text{LN}(\mathbf{H}^{(l-1)})) + \mathbf{H}^{(l-1)} \tag{5}$$

$$\mathbf{H}^{(l)} = \text{FFN}(\text{LN}(\hat{\mathbf{H}}^{(l-1)})) + \hat{\mathbf{H}}^{(l-1)}. \tag{6}$$

We apply the layer normalization (LN) before the multi-head self-attention (MHA) and the feed-forward network (FFN).

## 5.3 Optimization and Inference

In the training and inference, we sample $\mathcal{S}$ input sequences for each center node and use the center node representation from the final Transformer layer $z_c^{(s)}$ for prediction. Note that the computational complexity is controllable by choosing suitable number of sampled nodes. A MLP (Multi-Layer Perceptron) is used to predict the node class:

$$\widetilde{\boldsymbol{y}}^{(s)} = f_{MLP}\left(\boldsymbol{z}_c^{(s)}\right), \tag{7}$$

where $\widetilde{\boldsymbol{y}}_c \in \mathbb{R}^{C \times 1}$ stands for the classification result, $C$ stands for the number of classes. In the training process, we optimize the average cross entropy loss of labeled training nodes $V_L$:

$$\mathcal{L} = -\frac{1}{\mathcal{S}} \sum_{v_i \in V_L} \sum_{s=1}^{\mathcal{S}} \boldsymbol{y}_i^T \log \widetilde{\boldsymbol{y}}_i^{(s)}, \tag{8}$$

where $\boldsymbol{y}_i \in \mathbb{R}^{C \times 1}$ is the ground truth label of center node $v_i$. In the inference stage, we take a bagging aggregation to improve accuracy and reduce variance:

$$\widetilde{\boldsymbol{y}}_i = \frac{1}{\mathcal{S}} \sum_{s=1}^{\mathcal{S}} \widetilde{\boldsymbol{y}}_i^{(s)}. \tag{9}$$

Table 1: Node classification performance (mean±std%, the best results are bolded).

| Model | Cora | Citeseer | Pubmed | Chameleon | Actor | Squirrel | Texas | Cornell | Wisconsin |
|---|---|---|---|---|---|---|---|---|---|
| GCN | 87.33±0.38 | 79.43±0.26 | 84.86±0.19 | 60.96±0.75 | 31.39±0.23 | 43.15±0.18 | 75.16±0.95 | 66.74±1.39 | 64.31±2.16 |
| GAT | 86.29±0.53 | 80.13±0.62 | 84.40±0.05 | 63.90±0.46 | 36.05±0.35 | 42.72±0.39 | 78.76±0.87 | 76.04±1.35 | 66.01±3.48 |
| GraphSAGE | 86.90±0.84 | 79.23±0.53 | 86.19±0.18 | 62.15±0.42 | 38.55±0.46 | 41.26±0.26 | 79.03±1.44 | 72.54±1.50 | 79.41±3.60 |
| APPNP | 87.15±0.43 | 79.33±0.35 | 87.04±0.17 | 51.91±0.56 | 38.80±0.25 | 37.76±0.45 | 91.18±0.75 | 91.75±0.72 | 82.56±3.57 |
| JKNet | 87.70±0.65 | 78.43±0.31 | 87.64±0.26 | 62.92±0.49 | 33.42±0.28 | 42.60±0.50 | 77.51±1.72 | 64.26±1.16 | 81.20±1.96 |
| $H_2$GCN | 87.92±0.82 | 77.60±0.76 | 89.55±0.14 | 61.20±0.95 | 36.22±0.33 | 38.51±0.20 | 86.37±2.67 | 84.93±1.89 | 87.73±1.57 |
| GPRGNN | 88.27±0.40 | 78.46±0.88 | 89.38±0.43 | 64.56±0.59 | 39.27±0.21 | **46.34±0.77** | 91.84±1.25 | 90.25±1.93 | 86.58±2.58 |
| GT | 71.84±0.62 | 67.38±0.76 | 82.11±0.39 | 57.86±1.20 | 37.94±0.26 | 25.68±0.22 | 66.70±1.13 | 60.39±1.66 | 65.08±4.37 |
| SAN | 74.02±1.01 | 70.64±0.97 | 86.22±0.43 | 55.62±0.43 | 38.24±0.53 | 25.56±0.23 | 70.10±1.82 | 61.20±1.17 | 65.30±3.80 |
| Graphormer | 72.85±0.76 | 66.21±0.83 | 82.76±0.24 | 36.81±1.96 | 37.85±0.29 | 25.45±0.12 | 68.56±1.74 | 59.41±1.21 | 67.53±3.38 |
| Gophormer | 87.65±0.20 | 76.43±0.78 | 88.33±0.44 | 57.40±0.14 | 37.50±0.42 | 37.85±0.36 | 88.25±1.96 | 89.46±1.51 | 85.09±2.60 |
| ANS-GT | **88.60±0.45** | **80.25±0.39** | **89.56±0.55** | **65.42±0.71** | **40.10±1.12** | 45.88±0.34 | **93.24±1.85** | **92.10±1.78** | **88.62±2.24** |

## 5.4 Computational Complexity

Compared with existing Graph Transformers, ANS-GT requires extra computational costs on graph coarsening and sampling weights update. Here we want to show that the overall computational complexity of ANS-GT is linear with the number of nodes $n$. Hence, ANS-GT is scalable to large graphs. First, the computational complexity of graph coarsening is linear with $n$ [26] and we only need to do it once before training. Second, the computational cost of self-attention calculation in one epoch is $\mathcal{O}(n\mathcal{S}(N + n_s + n_g)^2)$ where the number of sampled nodes $N$, the number of sampled super-nodes $n_s$, the number of global nodes $n_g$, and the number of augmentations $\mathcal{S}$ are specified constants. Finally, the cost of updating sampling weights is linear to $n$, which is mainly attributed to the computation of rewards. Empirical efficiency analyses of ANS-GT are shown in the Appendix C.

## 6 Experiments

### 6.1 Experimental Setup

**Datasets.** To comprehensively evaluate the effectiveness of ANS-GT, we conduct experiments on the six benchmark datasets including citation graphs Cora, CiteSeer, and PubMed [18]; Wikipedia graphs Chameleon, Squirrel; the Actor co-occurrence graph [5]; and WebKB datasets [28] including Cornell, Texas, and Wisconsin. We set the train-validation-test split as 60%/20%/20%. The statistics of datasets are shown in the Appendix C.

**Baselines.** To evaluate the effectiveness of ANS-GT on graph representation learning, we compare it with 12 baseline methods, including 8 popular GNN methods, i.e. GCN [18], GAT [34], GraphSAGE [11], JKNet [36], APPNP [20], Geom-GCN [28], $H_2$GCN [48], and GPRGNN [6] along with four state-of-the-art Graph Transformers, i.e. GT [9], SAN [21], Graphormer [38], and Gophormer [46]. We use node classification accuracy as the evaluation metric.

**Implementation Details.** We adopt AdamW as the optimizer and set the hyper-parameter $\epsilon$ to 1e-8 and $(\beta 1, \beta 2)$ to (0.99,0.999). The peak learning rate is set to 2e-4 with a 100 epochs warm-up stage followed by a linear decay learning rate scheduler. We adopt the Variational Neighborhoods [26] with a coarsening rate of 0.01 as the default coarsening method. All models were trained on one NVIDIA Tesla V100 GPU.

**Parameter Settings.** In the default setting, the dropout rate is set to 0.5, the end learning rate is set to 1e-9, the hidden dimension $d$ is set to 128, the number of training epochs is set to 1,000, update period $T$ is set to 100, $N$ is set to 20, $M$ is set to 10, and the number of attention head $H$ is set as 8. We tune other hyper-parameters on each dataset based on by grid search. The searching space of batch size, number of data augmentation $\mathcal{S}$, the number of layers $L$, number of sampled nodes, number of sampled super-nodes, number global nodes are $\{8, 16, 32\}$, $\{4, 8, 16, 32\}$, $\{2, 3, 4, 5, 6\}$, $\{10, 15, 20, 25\}$, $\{0, 3, 6, 9\}$, $\{1, 2, 3\}$ respectively.

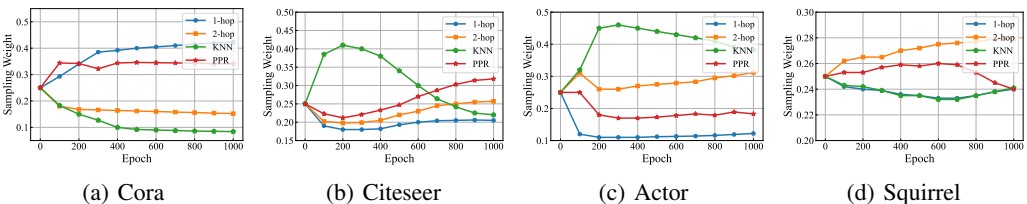

|   (a) Cora   |   (b) Citeseer   |   (c) Actor   |   (d) Squirrel   |

Figure 3: The normalized sampling weights as a function of training epochs on four datasets.

## 6.2 Effectiveness of ANS-GT

**Node Classification Performance.** The node classification results are shown in Table 1. We apply 3 independent runs on random data splitting and report the means and standard deviations. We have the following obervations: (1) Generally, we observe that ANS-GT overperforms all Graph Transformer baselines and achieves state-of-the-art results on nearly all datasets, which demonstrates the effectiveness of our proposed model. (2) We note that some Graph Transformer baselines achieve poor performance on node classification (e.g., GT only obtains $25.68\%$ on Squirrel) compared with graph neural network models. This is probably due to the full graph attention mechanisms or the fixed node sampling schemes of existing Graph Transformers. For instance, ANS-GT achieves an accuracy of $45.88\%$ on Squirrel while the best baseline has $43.15\%$.

**Effectiveness of Adaptive Node Sampling.** Our proposed adaptive node sampling module can adjust the weights for sampling based on the rewards as the training progresses. To evaluate its effectiveness and give more insights into the ANS module, we show the normalized sampling weights as a function of training epochs on four datasets in Figure 3. Generally, we observe that the sampling weights of different sampling strategies are time-sensitive and gradually stabilize with the increase of the number of epochs. Interestingly, we find PPR and 1-hop neighbors achieves high weights on Cora while 2-hop neighbors dominate other sampling strategies on Squirrel. This may be explained by the fact that Cora and Squirrel are strong homophily/heterophily dataset respectively. For Citeseer and Actor, the weights of KNN firstly goes up and gradually decreases. This is probably due to the reason that nodes with similar attributes are most useful for

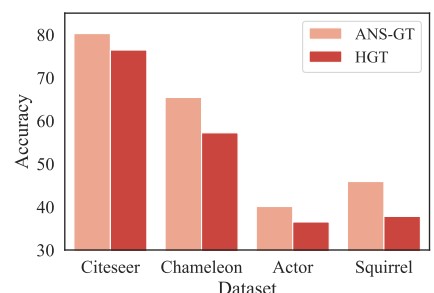

Figure 4: Ablation studies to show the effectiveness of the adaptive node sampling module. HGT refers to the Hierarchical Graph Transformer without adaptive node sampling.

the training at the beginning stage; local neighbors such as nodes with high PPR scores are more useful at the fine-tuning stage. Furthermore, Figure 4 shows the ablation studies of the adaptive node sampling module. We can observe that ANS-GT has a large advantage compared to its variant without the adaptive sampling module denoted as HGT (e.g., On Chameleon, ANS-GT achieves $65.42\%$ while HGT only has $57.20\%$).

**Graph Coarsening Methods.** In ANS-GT, we apply a hierarchical attention mechanism where the interactions between the center node and supernodes generated by graph coarsening are considered. Here, we evaluate ANS-GT with different coarsening algorithms and different coarsening rates. Specifically, the considered coarsening algorithms include Variation Neighborhoods (VN) [26], Variation Edges (VE) [26], and Algebraic (JC) [29]. We vary the coarsening rate from 0.01 to 0.50. In Table 2, we can observe that there is no significant difference between different coars-

Table 2: Sensitivity analysis of coarsening algorithms and coarsening rate.

| Dataset | Method | c=0.01 | c=0.05 | c=0.10 | c=0.50 | c=1.00 |
|---------|--------|--------|--------|--------|--------|--------|
| Cora    | VN     | **88.60** | **88.55** | 88.14  | **87.85** | 87.26  |
|         | VE     | 87.95  | 88.13  | **88.30** | 87.32  | 87.22  |
|         | JC     | 88.49  | 88.20  | 87.46  | 87.36  | **87.28** |
| Actor   | VN     | **39.72** | 39.45  | **40.10** | 38.83  | 39.08  |
|         | VE     | 39.20  | **39.66** | 39.51  | 38.94  | 39.06  |
|         | JC     | 39.15  | 39.85  | 39.92  | **39.16** | **39.09** |

ening algorithms, indicating the robustness of ANS-GT w.r.t. them. As for the coarsening rate, the results indicate that the coarsening rate of 0.01 to 0.10 has the best performance.

# 7    Conclusion

Motivated by the obstacles to generalize Transformer to large graphs, we propose **A**daptive **N**ode **S**ampling for **G**raph **T**ransformer (ANS-GT), which modifies a multi-armed bandit algorithm to adaptively sample nodes for attention in this paper. To incorporate long-range dependencies and global contexts, we further design a hierarchical graph attention scheme in which coarse-grained attention is achieved with graph coarsening. We empirically evaluate our method on six benchmark datasets to show the advantage over existing Graph Transformers and popular GNNs. The detailed analysis demonstrates that the adaptive node sampling module could effectively adjust the sampling strategies according to graph properties. Finally, We hope our work can help Transformer generalize to the graph domain and encourage the unified modeling of multi-modal data.

# 8    Acknowledgments

This research was partially supported by grants from the National Natural Science Foundation of China (Grant No.s 61922073 and U20A20229).

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
