## A  More Details of Motivating Observations

**Experiment Setup.** We conduct experiments on the Newman artificial networks [10] with different properties. The network consists of 128 nodes divided into 4 classes, where each node has on average $z_{in}$ edges (i.e., intra-class edges) connecting to nodes of the same class and $z_{out}$ edges (i.e., inter-class edges) to nodes of other classes, and $z_{in} + z_{out} = 16$. Here two indicators are used: $\rho_{in} = z_{in}/32$ and $\rho_{out} = z_{out}/96$, to indicate the graph property, i.e., $\rho_{in} > \rho_{out}, \rho_{in} = \rho_{out}$ and $\rho_{in} < \rho_{out}$ means the graph with homophily, randomness and heterophily, respectively. In Figure 5, we show the visualization of the adjacency matrix with strong homophily, randomness and strong heterophily.

For the node attributes, we generate $4h$-dimensional binary attributes (i.e., $x_i$) for each node to form 4 attribute clusters, corresponding to the 4 classes [13]. To be specific, for every node in the $i$-th class, we use a binomial distribution with mean $p_{in} = h_{in}/h$ to generate a $h$-dimensional binary vector as its $((i-1) \times h + 1)$-th to $(i \times h)$-th attributes, and generated the rest attributes using a binomial distribution with mean $p_{out} = h_{out}/(3h)$. In our experiments, we set $4h = 200$ and $h_{out} = 4(h_{in} + h_{out} = 16)$, so that $p_{in} > p_{out}$, the $h$-dimensional attributes are associated with the $i$-th class with a higher probability, whereas the rest $3h$ attributes are irrelevant. For the model implementation, we use the Gophormer [43] with the default setting for the demonstration. For each center node, we sample 10 nodes with 1-hop, 2-hop, KNN and PPR strategies 16 times for data augmentation.

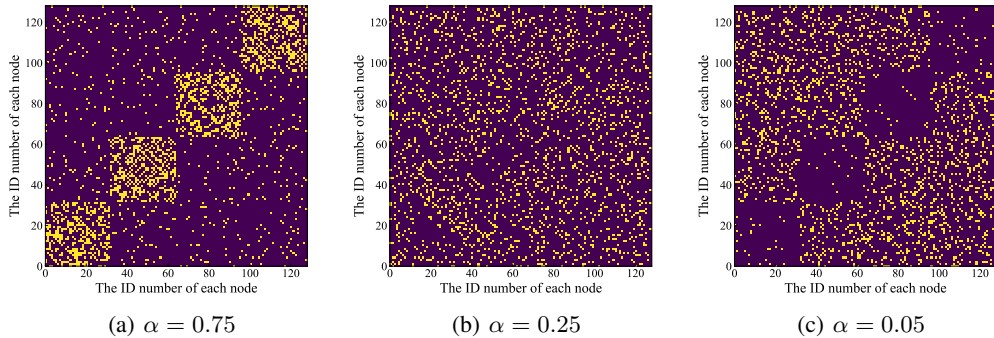

(a) $\alpha = 0.75$        (b) $\alpha = 0.25$        (c) $\alpha = 0.05$

Figure 5: The adjacency matrix of the Newman network with strong homophily, randomness and strong heterophily respectively. (The yellow dots indicate connected edges and the purple dots indicate no edges.)

## B  Dataset Statistics

In Table 3, we show the detailed statistics of 9 datasets.

Table 3: The statistics of the datasets.

| Dataset | #Nodes | #Edges | #Classes | #Features | Type | $\alpha$ |
|---------|--------|--------|----------|-----------|------|----------|
| Cora | 2,708 | 5,278 | 7 | 1,433 | Citation network | 0.83 |
| Citeseer | 3,327 | 4,522 | 6 | 3,703 | Citation network | 0.71 |
| Pubmed | 19,717 | 44,324 | 3 | 500 | Citation network | 0.79 |
| Chameleon | 2,277 | 31,371 | 5 | 2,325 | Wiki pages | 0.23 |
| Actor | 7,600 | 26,659 | 5 | 932 | Actors in movies | 0.22 |
| Squirrel | 5,201 | 198,353 | 5 | 2,089 | Wiki pages | 0.22 |
| Texas | 183 | 279 | 5 | 1703 | Web pages | 0.11 |
| Cornell | 183 | 277 | 5 | 1703 | Web pages | 0.30 |
| Wisconsin | 251 | 499 | 5 | 1703 | Web pages | 0.21 |

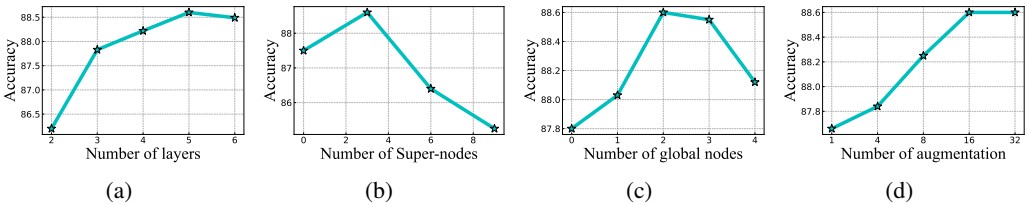

(a)         (b)         (c)         (d)

Figure 6: Parameter sensitivity analysis on Cora. We show (a) the influence of the number of layers; (b) the number of super-nodes; (c) the number of global nodes; (d) and the number of augmentation.

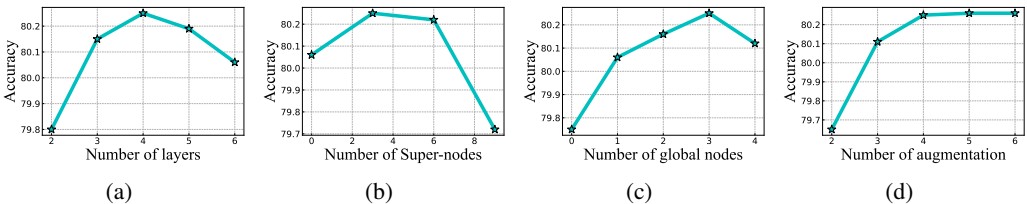

(a)         (b)         (c)         (d)

Figure 7: Parameter sensitivity analysis on Citeseer.

# C    Additional Results and Analysis

Table 4: Time consumption for graph coarsening (s). The coarsening rate $c$ is defined as $\frac{|V'|}{|V|}$

| Dataset | Method | c=0.01 | c=0.10 | c=0.50 |
|---------|--------|--------|--------|--------|
| Cora | VN | 3.792 | 3.536 | 2.215 |
| | VE | 1.540 | 1.516 | 0.851 |
| | JC | 1.454 | 1.271 | 0.665 |
| Actor | VN | 11.868 | 11.53 | 7.000 |
| | VE | 7.535 | 6.911 | 3.154 |
| | JC | 11.785 | 11.624 | 3.651 |

## C.1   Hyper-parameter Analysis

In Figure 6, we study the sensitivity of ANS-GT on four important hyper-parameters: the number of transformer layers, the number of super-nodes $n_s$, the number of global nodes $n_g$, and the number of data augmentation $\mathcal{S}$. In Figure 6 (a), we observe that the performance increases at the beginning with the increase of transformer layers. The reason is that stacking more transformer layers improves the model's capability. However, we witness a slight performance decrease when the number of layers exceeds 6, possibly suffering from over-fitting. Figure 6 (b) and (c) presents the node classification performance with $n_s$ varying from 0 to 9 and $n_g$ from 0 to 4 respectively. With the increase of $n_s$ and $n_g$, the performance increases until reaches a peak and then decreases. This is expected as the optimal number of super-nodes and global nodes help incorporate long-range dependencies and global context in the graph while too large $n_s$ and $n_g$ lead to redundant noise. Hence, the number of super-nodes and global nodes should be carefully chosen to achieve optimal performance. Finally, we show the influence of the number of data augmentation in Figure 6 (d). With the increase of $\mathcal{S}$, the node classification performance improves steadily until stabilizes. The results indicate data augmentation in the training and the bagging aggregation in the inference can effectively improve the classification accuracy. In conclusion, we recommend 5 transformer layers, 3 super-nodes, 2 global nodes, and an augmentation number of 16 for Cora.

## C.2   Efficiency Analysis of ANS-GT

Here we show more experiment results and analysis on the efficiency of ANS-GT.

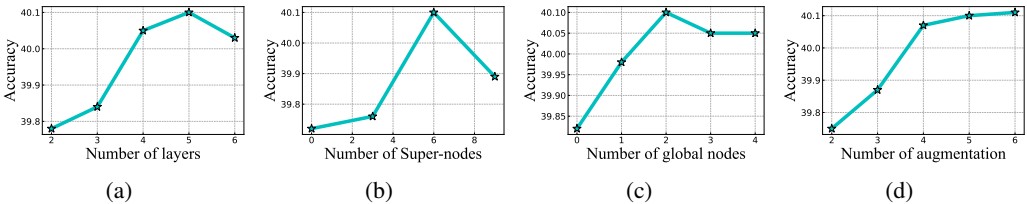

|  | (a) | (b) | (c) | (d) |

Figure 8: Parameter sensitivity analysis on Actor.

Table 5: Time consumption for adaptive node sampling per epoch (s).

| Dataset | Cora | Citeseer | Pubmed | Chameleon | Actor | Squirrel |
|---|---|---|---|---|---|---|
| Time | 0.838 | 0.926 | 1.717 | 0.855 | 1.026 | 0.977 |

In Table 4, we present the time consumption of executing the graph coarsening algorithm on Cora and Actor with different coarsening rates and methods. Since graph coarsening only needs to be done once at the pre-processing stage, the time consumption is acceptable.

In Table 5, we show the time consumption for adaptive sampling in one epoch. In our algorithm, we update the sampling weights every $T$ epochs ($T = 100$ in experiments). Hence, the time cost of the adaptive node sampling module is trivial.

In Table 6, we present the training efficiency comparisons with other Graph Transformer baselines. Specifically, we show the average training time per epoch. As can be observed in Table 6, ANS-GT has comparable training time with Gophormer and its efficiency is much better than Graphormer.

### C.3 Limitations and Potential Negative Social Impacts

One limitation of our work is that it introduces more hyper-parameters for finetuning. Since our work utilizes adaptive node sampling, it may lead to potential biases in sampling nodes for training.

### C.4 Additional Results on OGB Datasets

We additionally try ANS-GT on ogbn-arxiv and ogbn-products datasets from OGB [15], which contains 169,343 and 2,449,029 nodes respectively. We use the official train/valid/test split and data pre-processing details can be found in [15]. The model setup of ANS-GT follows Section 6.1. Three competitive baselines including GCN, GraphSAGE, and GPRGNN are selected. We present average accuracies and standard deviations over 5 runs in Table 7. Our results overperform the baselines and demonstrate the effectiveness of ANS-GT on large-scale graphs.

## D   Supplementary Information of Graph Coarsening

In this paper, we use 3 popular graph coarsening algorithms: Variation Neighborhoods (VN) [26], Variation Edges (VE) [26], and Algebraic JC (JC) [30]. VN and VE belong to the local variation algorithms which coarsen graphs by preserving the spectral properties of adjacency matrix. Local variation algorithms differ only in the type of contraction sets that they consider: Variation Edges only contracts edges, whereas contraction sets in Variation Neighborhoods are subsets of nodes' neighborhood. In Algebraic JC, the algebraic distances between neighboring nodes are calculated and close nodes are contracted to form clusters. More information of the coarsening algorithms can be found in their original papers.

## E   Further Discussions of ANS-GT

In ANS-GT, we formulate the optimization strategy of node sampling in Graph Transformer as an adversary bandit problem. Specifically, ANS-GT optimizes the weights of chosen sampling heuristics instead of directly predicting the adjacent nodes to attend. Then, ANS-GT combines the weighted

Table 6: Efficiency comparisons with Graph Transfomer baselines. The average training time per epoch (s)

| Dataset | Cora | Citeseer | Pubmed | Chameleon | Actor | Squirrel |
|---|---|---|---|---|---|---|
| Graphormer | 25.670 | 37.899 | 26.436 | 26.343 | 30.105 | 29.771 |
| Gophormer | 11.210 | 12.121 | 16.116 | 10.305 | 12.243 | 12.579 |
| ANS-GT | 11.495 | 12.143 | 16.270 | 10.311 | 12.240 | 12.571 |

Table 7: The performance of ANS-GT and selected baselines on OGB datasets

| Methods | GCN | GraphSAGE | GPRGNN | ANS-GT |
|---|---|---|---|---|
| ogbn-arxiv | 71.72±0.45 | 71.46±0.26 | 70.90±0.23 | **72.84±0.34** |
| ogbn-products | 75.57±0.28 | 78.61±0.31 | 79.76±0.59 | **82.15±0.30** |

sampling heuristics to sample informative nodes. We did not incorporate hierarchical attention as part of the bandit learning because it samples supernodes from the coarsened graph instead of sampling nodes like the 4 strategies (1-/2- hops, KNN, and PPR). We do not directly predict nodes to attend (e.g., using linear layers to predict informative nodes). Directly predicting informative nodes for attention requires too much computational overhead and is hard to optimize. Comparatively, the pre-defined node sampling heuristics in our strategy help narrow the search space with prior knowledge. Moreover, the sampling strategy in ANS-GT can generalize to all nodes in the graph efficiently. Experiment results show that our strategy is effective and efficient.