# OpenReview forum: "Hierarchical Graph Transformer with Adaptive Node Sampling"
_NeurIPS.cc/2022/Conference — NeurIPS 2022 Accept_

### Official Review · Reviewer_6Yic · 2022-07-10

**Rating:** 7
**Confidence:** 3
**Soundness:** 4 excellent
**Presentation:** 4 excellent
**Contribution:** 4 excellent

**Summary:**

The paper tries to scale up graph transformers to larger graphs with two novel methods. They first identify that the challenge of large graphs for graph transformers are agnostic about the graph structure during training, which motivates them to use bandit learning to utilize multiple node sampling strategies simultaneously. Another challenge is that the graph transformers are unable to capture distant information, which was tackled by introducing a hierarchical attention mechanism that allows the nodes to attend to a coarsened graph sparsely. While keeping the computational complexity linear to the number of nodes, the proposed method achieves state-of-the-art results on multiple datasets.

**Questions:**

1. Although the hierarchical graph attention is a novel method proposed in this paper and the other 4 strategies (1-/2- hops, KNN, and PPR) are existing ones, all of them can be treated as a distribution over all nodes. In this sense, I wonder why don't you incorporate hierarchical attention as part of the bandit learning?

2. For the attention over the coarsened graph, how do you represent the supernodes? Do you aggregate the node features within each cluster, or do you pick a representative? Or do they have their own features that are not shared with their member nodes (if so, how are they related to their members?)?

**Limitations:**

I don't think this paper will have potential societal impacts.

**Strengths And Weaknesses:**

The paper is well-motivated, with an investigation of current node sampling strategies to show that the effective sampling strategies may depend on the actual dataset. The proposed solution, i.e. adaptive sampling with bandit learning, well fits the needings and is empirically supported by the evidence shown in fig 3. The other improvement -- the hierarchical attention -- is both novel and effective for distant information capturing.

The paper is well written, with a clear structure and simple notations to follow. Together with the diagrams shown in fig 2, the presentation of this work is neat and illustrative.

The experiment results are strong, with consistent improvement over the baselines. The ablation study shows that both tricks are working and effective, especially compared to other graph transformers.

The paper is great overall, and I trust the paper would have a positive impact on the field of GNNs. However, I also observe a few drawbacks that might hurt the influence of the paper:

1. Using bandit learning to tune the weights of sample strategies enables the model to automatically adjust the weights, and fig 3 shows that the weights would quickly converge. However, it is not convincing to show the necessity of bandit learning. I wonder if a baseline that simply takes the average of 4 distributions (i.e. fixing the weights to be 0.25) can achieve similar performance.

2. While the coarsened graph is a good choice to capture distant information, it introduces extra algorithmic computational overhead. It would be good if the authors can provide an estimate of the running time of the coarsening algorithm compared to the training/inference of the graph models.

3. A minor suggestion: It is a bit confusing to boldface "Reward Scheme" in L226 because you just said you would use 4 sampling approaches and have exactly 4 boldfaced items.

4. It would be good to provide more details about the employment of the coarsening algorithm. A succinct description of the algorithm will make the paper more self-contained.

---

> ### Author Response · Authors · 2022-08-01
> **Response to Reviewer 6Yic**
>
> We thank the reviewer for the detailed and valuable suggestions. Please see below for detailed responses for the comments.
>
> **Comment1:** Using bandit learning to tune the weights of sample strategies enables the model to automatically adjust the weights, and fig 3 shows that the weights would quickly converge. However, it is not convincing to show the necessity of bandit learning. I wonder if a baseline that simply takes the average of 4 distributions (i.e. fixing the weights to be 0.25) can achieve similar performance.
>
> **Response1:** We thank the reviewer for the insightful question. We did ablation studies by fixing the sampling weights to 0.25. The experimental setups are the same as the settings in Section 6.1. In the following table, we can observe that ANS-GT with adaptive node sampling clearly overperforms ANS-GT with fixing sampling weights, which shows the necessity of bandit learning.
>
> | Datasets                        | Cora  | Citeseer | Pubmed | Chameleon | Actor | Squirrel |
> | ------------------------------- | ----- | -------- | ------ | --------- | ----- | -------- |
> | ANS-GT(fixing sampling weights) | 87.31 | 78.32    | 88.13  | 61.20     | 38.56 | 40.17    |
> | ANS-GT                          | **88.60** | **80.25**    | **89.56**  | **65.42**     | **40.10** | **45.88**    |
>
> **Comment2:** While the coarsened graph is a good choice to capture distant information, it introduces extra algorithmic computational overhead. It would be good if the authors can provide an estimate of the running time of the coarsening algorithm compared to the training/inference of the graph models.
>
> **Response2:** We agree that the coarsened graph introduces extra computational overhead. In Appendix C.2, we present the computational time for executing the graph coarsening algorithm with different coarsening rates and methods. Since graph coarsening only needs to be done once at the pre-processing stage, we find that the computational time is acceptable.
>
> **Comment3:** A minor suggestion: It is a bit confusing to boldface "Reward Scheme" in L226 because you just said you would use 4 sampling approaches and have exactly 4 boldfaced items.
>
> **Response3:** Thank you for the valuable suggestion. We removed the boldface "Reward Scheme" in our revised version.
>
> **Comment4:** It would be good to provide more details about the employment of the coarsening algorithm. A succinct description of the algorithm will make the paper more self-contained.
>
> **Response4:** We thank the reviewer for the constructive suggestion. In this paper, we use 3 popular graph coarsening algorithms: Variation Neighborhoods (VN), Variation Edges (VE), and Algebraic JC (JC). VN and VE belong to the local variation algorithms which coarsen graphs by preserving the spectral properties of adjacency matrix. Local variation algorithms differ only in the type of contraction sets that they consider: Variation Edges only contracts edges, whereas contraction sets in Variation Neighborhoods are subsets of nodes’ neighborhood. In Algebraic JC, the algebraic distances between neighboring nodes are calculated and close nodes are contracted to form clusters. In the revised paper, we add the description of the coarsening algorithms in Appendix.
>
> **Comment5:** Although the hierarchical graph attention is a novel method proposed in this paper and the other 4 strategies (1-/2- hops, KNN, and PPR) are existing ones, all of them can be treated as a distribution over all nodes. In this sense, I wonder why don't you incorporate hierarchical attention as part of the bandit learning?
>
> **Response5:** We did not incorporate hierarchical attention as part of the bandit learning because it samples supernodes from the coarsened graph instead of sampling nodes like the 4 strategies (1-/2- hops, KNN, and PPR). Thus, it may be difficult to optimize them together in the bandit learning.
>
> **Comment6:** For the attention over the coarsened graph, how do you represent the supernodes? Do you aggregate the node features within each cluster, or do you pick a representative? Or do they have their own features that are not shared with their member nodes (if so, how are they related to their members?)?
>
> **Response6:** We aggregate the node features within each cluster to represent the supernodes. Specifically, the feature matrix of the coarsened graph $G'$ is defined by $X'\triangleq P^TX$, where $P$ is the normalized partition matrix and $X$ is the original feature matrix. More details can be found in Section 3.3 in the revised paper.

---

> > ### Comment · Reviewer_6Yic · 2022-08-08
> > **Response**
> >
> > Thanks for the authors to answer my questions. They are clear and helpful. The authors should consider adding them to the next revision of the paper.

---

### Official Review · Reviewer_BtvG · 2022-07-10

**Rating:** 6
**Confidence:** 5
**Soundness:** 4 excellent
**Presentation:** 3 good
**Contribution:** 3 good

**Summary:**

This paper proposes a new transformer-based architecture for graph-structured data with adaptive node sampling and hierarchical attention. To learn effective sparse token/node sampling, the authors cast the learning as a multi-arm bandit problem to capture long-range dependency as well as messages from close neighbors. Unlike previous sampling strategies that do not take into account the characteristics of graphs, the proposed method learns the effective sampling by adjusting the weights of a set of sampling schemes. Optimizing these weights is formulated as a bandit problem.

**Questions:**

* Transformer with multi-arm bandit formulation is studied in the context of memory replay. Transformer with Memory Replay [AAAI2022]: Transformer with Token Replay. It is not directly related to graph neural networks, but more transformer+multi-arm bandit frameworks could be discussed in the related work section.
* Heterophily graph data have been recently studied by multiple seminal works. Geom-GCN, H_2GCN, and more recent works. The previous works compared their performance with more datasets: Texas, Wisconsin, and Cornell. Also, H2GCN achieved competitive performance. Those methods must be included as baselines. Another problem is that the dataset split is different from the graph neural networks which studied heterophilic graphs. Because of that, the performance provided in this paper is not compatible with GNNs, which are specifically designed for heterophilic graphs.
* If the attention is used to define the reward, then why are the pre-defined node sampling methods needed? Is there a way to directly predict the nodes that will get the high attention scores and use those as a distribution for sampling?
* Do the node sampling methods need to provide relative importance of nodes given a centner node? How can you update the relative importance of other nodes given a center node? In addition, if the attention was computed for each center node, then why does attention needs to be computed for all tokens as in Figure 2? Figure 2 was a bit busy and confusing.

**Limitations:**

* Overall, the paper is well-written. The proposed method has many interesting components, and it is a novel construction. However, the experiments of this paper failed to include strong and recent graph neural networks, which have shown their effectiveness on both homophilic and heterophilic graphs. Also, more heterophilic graphs could be considered as previous works in the literature.
* Node sampling is the main contribution, but the discussion about supernodes and hierarchical attention was relatively less clear due to a short introduction.

**Strengths And Weaknesses:**

* This paper pinpoints the limitations of transformers for graph-structured data. Especially for large-scale graphs, transformers suffer from quadratic scalability. Although hierarchical transformers with windows like SWIN transformer have been studied, due to the irregularity of graphs, they could not be directly extended to the graph domain. So, combining with previous efforts to design efficient node sampling schemes in the graph domain, the authors proposed a strategy to learn to combine multiple node sampling schemes. It is an interesting direction.
* The paper is well-written. Even though this paper requires knowledge of multiple previous works, this paper concisely introduced related methods with equations.
* The proposed method achieved significant improvement on top of a Graph Transformer.

---

> ### Author Response · Authors · 2022-08-01
> **Response to Reviewer BtvG (1/2)**
>
> We appreciate the constructive comments for our work, which will help us improve our work in the future. Please see below for detailed responses for the comments.
>
> **Comment1:** Transformer with multi-arm bandit formulation is studied in the context of memory replay. Transformer with Memory Replay [AAAI2022]: Transformer with Token Replay. It is not directly related to graph neural networks, but more transformer+multi-arm bandit frameworks could be discussed in the related work section.
>
> **Response1:** Thanks for mentioning the related work. We will discuss more related work on transformer+multi-arm bandit including this AAAI2022 paper in our future version.
>
> **Comment2:** Heterophily graph data have been recently studied by multiple seminal works. Geom-GCN, H_2GCN, and more recent works. The previous works compared their performance with more datasets: Texas, Wisconsin, and Cornell. Also, H2GCN achieved competitive performance. Those methods must be included as baselines.
> Another problem is that the dataset split is different from the graph neural networks which studied heterophilic graphs. Because of that, the performance provided in this paper is not compatible with GNNs, which are specifically designed for heterophilic graphs.
>
> **Response2:** We thank the reviewer for mentioning the related works on Heterophily graph data. Below we compare Geom-GCN, H_2GCN, and GPR-GNN with ANS-GT on 9 datasets. Our dataset split is consistent with that specified in Geom-GCN. We report the results of Geom-GCN from [1] and run H_2GCN[2] and GPRGNN[3] with the same dataset split for a fair comparison. We use the code from their original papers and report the average accuracy on the test sets over 5 independent runs. We can observe that these heterophilic GNNs are indeed strong baselines while ANS-GT still achieved the best performance on 7 out of 9 datasets.
>
> | Datasets | Cora  | CiteSeer | PubMed | Chameleon | Actor | Squirrel | Texas | Cornell | Wisconsin
> | -------- | ----- | -------- | ------ | --------- | ----- | --------- | ----- | -------- | ----- |
> | Geom-GCN | 85.27 | 77.99    | **90.05**  | 60.90     | 31.63 | 38.14    | 67.57 | 60.81   | 64.12
> | H_2GCN   | 87.92 | 77.60    | 89.55  |  61.20     | 36.22 | 38.51    | 86.37 | 84.93   | 87.73
> | GPRGNN   | 88.27 | 78.46    | 89.38  |  64.56     | 39.27 | **46.34**    | 91.84 | 90.25   | 86.58
> | ANS-GT   | **88.60** | **80.25**    | 89.56  | **65.42**     | **40.10** | 45.88    | **93.24** | **92.10**   | **88.62**
>
> [1] Pei, Hongbin, et al. Geom-GCN: Geometric Graph Convolutional Networks. In ICLR, 2019.
>
> [2] Zhu, Jiong, et al. Beyond homophily in graph neural networks: Current limitations and effective designs. In NeurIPS 2020.
>
> [3] Eli Chien, et al. Adaptive universal generalized PageRank graph neural network. In ICLR, 2021.
>
> **Comment3:** If the attention is used to define the reward, then why are the pre-defined node sampling methods needed? Is there a way to directly predict the nodes that will get the high attention scores and use those as a distribution for sampling?
>
> **Response3:** In ANS-GT, we formulate the optimization strategy of node sampling in Graph Transformer as an adversary bandit problem. Specifically, ANS-GT optimizes the weights of chosen sampling heuristics instead of directly predicting the nodes to attend. Then, ANS-GT combines the weighted sampling heuristics to sample informative nodes.
> Directly predicting informative nodes for attention requires too much computational overhead and is hard to optimize. Comparatively, the pre-defined node sampling heuristics in our strategy help narrow the search space with prior knowledge. Moreover, the sampling strategy in ANS-GT can generalize to all nodes in the graph efficiently. Experiment results show that our strategy is effective and efficient.

---

> > ### Author Response · Authors · 2022-08-01
> > **Response to Reviewer BtvG (2/2)**
> >
> > **Comment4:** Do the node sampling methods need to provide relative importance of nodes given a centner node? How can you update the relative importance of other nodes given a center node? In addition, if the attention was computed for each center node, then why does attention needs to be computed for all tokens as in Figure 2? Figure 2 was a bit busy and confusing.
> >
> > **Response4:** Yes, the node sampling methods provide the relative importance of nodes given a center node. We show how to update the sampling strategy in Figure 2 and Algorithm 1. Generally, we update the weights of sampling heuristics based on the attention scores and combine these sampling heuristics to sample informative nodes.
> > In ANS-GT, the attention is computed only for the sampled nodes. We thank the reviewer for the suggestion and will make our figures clearer and easier to follow.
> >
> > **Comment5:** Node sampling is the main contribution, but the discussion about supernodes and hierarchical attention was relatively less clear due to a short introduction.
> >
> > **Response5:** Due to the page limits, we focus on the illustration of adaptive node sampling in the main text. We provided a discussion on the ablation studies of supernodes, the efficiency of graph coarsening, and hierarchical attention in the appendix. We will provide more discussions and experiments in the revised version.

---

> > > ### Comment · Reviewer_BtvG · 2022-08-08
> > > **I maintain my original rating.**
> > >
> > > I appreciate the authors for the detailed responses. I believe that the paper will be more comprehensive and solid if the answers are properly included in the final version. All my concerns are addressed. I maintain my original rating.

---

### Official Review · Reviewer_cmdK · 2022-07-11

**Rating:** 4
**Confidence:** 5
**Soundness:** 2 fair
**Presentation:** 3 good
**Contribution:** 3 good

**Summary:**

The paper proposes a Transformer architecture for graph-structured data, especially on large graphs. Most existing graph Transformers focus on small-scale graphs and have difficulty learning node representations on large-scale graphs due to a scalability issue. To generalize a Transformer to large graphs, most works use fixed node sampling strategies. This paper addresses these challenges by proposing Adaptive Node Sampling for Graph Transformer (ANS-GT), which is based on an adversary bandit problem. Also, to capture long-distance dependencies, the authors present a hierarchical attention scheme for Graph Transformer with graph coarsening algorithms.

**Questions:**


1. It would be better if the comparison with heterophilic graph neural networks is included. Is there any comparisons with heterophilic graph neural networks such as H2GCN and GPRGNN?
2. The contribution of each component of ANS-GT is unclear. Is there any ablation study on the graph coarsening and the global nodes?
3. ANS-GT is the graph Transformer for large-scale graphs. But the experiments has been conducted on relatively small datasets compared to OGB. If it is applicable, please include the experimental results on OGB datasets such as ogbn-arxiv.
4.  I think that the baselines except for Gophormer are trained without augmentation. So, I wonder the performance of ANS-GT when the number of input sequences $\mathcal{S}$ is 1.

**Limitations:**

The authors have adequately addressed the limitations.

**Strengths And Weaknesses:**

**Strengths**

1. The paper is clearly written and easy to follow.
2. The paper addresses the challenges of graph Transformers on large-scale graphs, which is an important problem.
3. The application of a multi-armed bandit mechanism to sampling informative nodes is interesting and seems clever.
4. Motivation is well articulated with experiments.

**Weakness**

1. The proposed methods consider heterophilic graphs. So, it would be better if graph neural networks for heterophilic graphs are included in related works. (Zhu, Jiong, et al., Chien, Eli, et al.)
2. There are many graph neural networks for heterophilic graph datasets such as H2GCN and GPRGNN (Zhu, Jiong, et al., Chien, Eli, et al.). It would be better if more baselines are included in Table1.
3. The paper proposes ANS-GT for generalizing graph Transformers to large-scale graphs. But, the largest dataset used in this paper is Pubmed, which consists of 19,717 nodes. At least I’d love to see the performance on OGB datasets such as ogbn-arxiv.
4. The contribution of each component of ANS-GT such as super nodes and global nodes is unclear. Even though the supplement includes ablation studies on the number of super nodes and global nodes, the experiments are limited to Cora dataset.
5. I think that the number of augmentation $\mathcal{S}$ needs to be considered when calculating the computational complexity of ANS-GT.

___
Zhu, Jiong, et al. "Beyond homophily in graph neural networks: Current limitations and effective designs.", NeurIPS 2020.

Chien, Eli, et al. "Adaptive universal generalized pagerank graph neural network.", ICLR 2021.

---

> ### Author Response · Authors · 2022-08-01
> **Response to Reviewer cmdK**
>
> We thank the reviewers for their insightful feedback. The reviewer asks great questions, and we provide the answers below.
>
> **Comment1:** The proposed methods consider heterophilic graphs. So, it would be better if graph neural networks for heterophilic graphs are included in related works.  There are many graph neural networks for heterophilic graph datasets such as H2GCN and GPRGNN (Zhu, Jiong, et al., Chien, Eli, et al.). It would be better if more baselines are included in Table1.
>
> **Response1:** We thank the reviewer for mentioning the related works on graph neural networks for heterophilic graphs. We will include these papers in related works.  Below we compare Geom-GCN, H_2GCN, and GPR-GNN with ANS-GT on 9 datasets (three extra heterophilic datasets). Our dataset split is consistent with that specified in Geom-GCN. We report the results of Geom-GCN from [1] and run H_2GCN[2] and GPRGNN[3] with the same dataset split for a fair comparison. We use the code from their original papers and report the average accuracy on the test sets over 5 independent runs. We can observe that these heterophilic GNNs are indeed strong baselines while ANS-GT still achieved the best performance on 7 out of 9 datasets. The best results are bolded.
>
> | Datasets | Cora  | CiteSeer | PubMed | Chameleon | Actor | Squirrel | Texas | Cornell | Wisconsin
> | -------- | ----- | -------- | ------ | --------- | ----- | --------- | ----- | -------- | ----- |
> | Geom-GCN | 85.27 | 77.99    | **90.05**  | 60.90     | 31.63 | 38.14    | 67.57 | 60.81   | 64.12
> | H_2GCN   | 87.92 | 77.60    | 89.55  |  61.20     | 36.22 | 38.51    | 86.37 | 84.93   | 87.73
> | GPRGNN   | 88.27 | 78.46    | 89.38  |  64.56     | 39.27 | **46.34**    | 91.84 | 90.25   | 86.58
> | ANS-GT   | **88.60** | **80.25**    | 89.56  | **65.42**     | **40.10** | 45.88    | **93.24** | **92.10**   | **88.62**
>
> [1] Pei, Hongbin, et al. Geom-GCN: Geometric Graph Convolutional Networks. In ICLR, 2019.
>
> [2] Zhu, Jiong, et al. Beyond homophily in graph neural networks: Current limitations and effective designs. In NeurIPS 2020.
>
> [3] Eli Chien, et al. Adaptive universal generalized PageRank graph neural network. In ICLR, 2021.
>
> **Comment2:** The paper proposes ANS-GT for generalizing graph Transformers to large-scale graphs. But, the largest dataset used in this paper is Pubmed, which consists of 19,717 nodes. I’d love to see the performance on OGB datasets such as ogbn-arxiv.
>
> **Response2:** We thank the reviewer for the constructive suggestion. We tried ANS-GT on ogbn-arxiv and ogbn-products datasets from OGB, which contains 169,343 and 2,449,029 nodes respectively. We use the official train/valid/test split and data pre-processing details can be found in [4]. The model setup of ANS-GT follows Section 6.1. Three competitive baselines including GCN, GraphSAGE, and GPRGNN are selected. We present average accuracies and standard deviations over 5 runs below. Our results overperform the baselines and demonstrate the effectiveness of ANS-GT on large-scale graphs.
>
> | Methods    | GCN        | GraphSAGE  | GPRGNN     | ANS-GT     |
> | ---------- | ---------- | ---------- | ---------- | ---------- |
> | ogbn-arxiv | 71.72±0.45 | 71.46±0.26 | 70.90±0.23 | **72.84±0.34** |
> | ogbn-products | 75.57±0.28 | 78.61±0.31 | 79.76±0.59 | **82.15±0.30** |
>
> Due to the time limit in the rebuttal period, we plan to include more results on OGB datasets in the future version.
>
> [4] Hu et al. Open graph benchmark: Datasets for machine learning on graphs. In NeurIPS, 2020
>
> **Comment3:** The contribution of each component of ANS-GT such as super nodes and global nodes is unclear. Even though the supplement includes ablation studies on the number of super nodes and global nodes, the experiments are limited to Cora dataset.
>
> **Response3:** In the supplementary, we provide more ablation studies on another two representative datasets (Citeseer and Actor) to show the contribution of super nodes and global nodes. ANS-GT has the best performance with an appropriate number of super nodes and global nodes. More analysis can be found in Appendix C.1. We will make the contribution of each component in ANS-GT clearer in the future version.
>
> **Comment4:** I think that the number of augmentation S needs to be considered when calculating the computational complexity of ANS-GT.
>
> **Response4:** Thanks for the valuable suggestion. We agree that the number of augmentation S needs to be considered when calculating the computational complexity of ANS-GT. We have revised Section 5.4 accordingly.
>
> **Comment5:** I think that the baselines except for Gophormer are trained without augmentation. So, I wonder the performance of ANS-GT when the number of input sequences S is 1.
>
> **Response5:** We have shown the performance of ANS-GT when the number of input sequences S is 1 in Appendix Figure 6(d),7(d), and 8(d). Generally, the performance of ANS-GT is sub-optimal when the number of augmentations is 1.

---

### Official Review · Reviewer_vEEU · 2022-07-15

**Rating:** 5
**Confidence:** 4
**Soundness:** 3 good
**Presentation:** 3 good
**Contribution:** 3 good

**Summary:**

Most existing node sampling strategies in Graph Transformers are agnostic to the graph properties and do not consider long-range relationships between nodes. To address this issue, the authors of the paper propose  Adaptive Node Sampling for Graph Transformer (ANS-GT), which includes an adaptive node sampling strategy to account for graph properties to improve the efficiency of the Graph Transformer. ANS-GT also incorporates complementary global information through graph coarsening and hierarchical attention schemes. State-of-the-art results are shown on 6 benchmark datasets for the semi-supervised node classification task.

**Questions:**

1. What is the difference between ANS-GT and models that learn linear layers to predict adjacent nodes to attend?

**Limitations:**

1. Mathematical formulas are confusing and a bit difficult to read.
2. The performance relies on the external graph coarsening method which is not updated during ANS-GT training.
3. It would be nice to see evaluations on graph tasks other than node classification.

**Strengths And Weaknesses:**

Strengths:
1. The authors provide sound statistics showing that no sampling strategy performs consistently for graphs with different homophily ratios, thus requiring an adaptive node sampling strategy.
2. They formulate the problem as a multi-armed bandit problem and use the attention score as a reward.

Weakness:
1. On line 133, bold uppercase H is used to denote the hidden state of the self-attention module, and bold lowercase h is used to denote the hidden state at position i. On line 137, an uppercase H is used to indicate the total number of heads, and a lowercase h is used to indicate heads. Also, on line 136, a bold uppercase W is used to denote the weight matrix in the self-attention module. On line 150, a capital W is used for edge weights. These letters are a bit overused and cause confusion. Sometimes bold uppercase letters are used for matrices, and sometimes lowercase letters are used for matrices. Sticking to one style would be great.
2. In Table 2 where different coarsening rates are compared, it'd be great to add c = 0 as a baseline and more possible rates.

---

> ### Author Response · Authors · 2022-08-01
> **Response to Reviewer vEEU**
>
> We thank the reviewer for the valuable comments and for pointing out parts that remain unclear. We have revised our paper following the suggestions and addressed all of your comments in the following response:
>
> **Comment1:** Mathematical formulas are confusing and a bit difficult to read. On line 133, bold uppercase H is used to denote the hidden state of the self-attention module, and bold lowercase h is used to denote the hidden state at position i. On line 137, an uppercase H is used to indicate the total number of heads, and a lowercase h is used to indicate heads. Also, on line 136, a bold uppercase W is used to denote the weight matrix in the self-attention module. On line 150, a capital W is used for edge weights. These letters are a bit overused and cause confusion. Sometimes bold uppercase letters are used for matrices, and sometimes lowercase letters are used for matrices. Sticking to one style would be great.
>
> **Response1:** We thank the reviewer for pointing out that some mathematical formulas are confusing. In the original submission, we use bold uppercase H: $\mathbf{H}$ and bold lowercase h: $\mathbf{h}$ to denote hidden states and use italic unbold H: $H$ and h: $h$ to indicate attention heads. Bold uppercase W: $\mathbf{W}$ is used to denote the weight matrix and italic unbold W: $W$ is used for edge weights.
>
> In the revised version, we use different symbols to make the mathematical formulas clearer. Specifically, we use uppercase B: $B$ to denote the total number of attention heads and use $A’$ to denote the weighted adjacency matrix of the coarsened graph $G’$.
>
> **Comment2:** In Table 2 where different coarsening rates are compared, it'd be great to add c = 0 as a baseline and more possible rates.
>
> **Response2:** In graph coarsening, the coarsening rate is defined as the ratio between the number of nodes in the coarse graph and the original graph. We guess the reviewer wants us to show c=1.0 as a straightforward baseline. In the following table, we add results of c=0.05 and c=1.0 based on Table 2. The results indicate that the coarsening rate of 0.01 to 0.10 has the best performance, which is consistent with our original observation.
>
> | Dataset | Method | c=0.01 | c=0.05 | c=0.1  | c=0.5  | c=1.0 |
> | ------- | ------ | ------ | ------ | ------ | ------ | ----- |
> | Cora    | VN     | 88.60  | 88.55  | 88.14  | 87.85  | 87.26 |
> |         | VE     | 87.95  | 88.13  | 88.30  | 87.32  | 87.22 |
> |         | JC     |  88.49 | 88.20  | 87.46  |  87.36 | 87.28 |
> | Actor   | VN     | 39.72  | 39.45  | 40.10  | 38.83  | 39.08 |
> |         | VE     |  39.20 | 39.66  |  39.51 |  38.94 | 39.06|
> |         | JC     | 39.15  | 39.85  | 39.92  | 39.16  | 39.09 |
>
> **Comment3:**  What is the difference between ANS-GT and models that learn linear layers to predict adjacent nodes to attend?
>
> **Response3:** In ANS-GT, we formulate the optimization strategy of node sampling in Graph Transformer as an adversary bandit problem. Specifically, ANS-GT optimizes the weights of chosen sampling heuristics instead of directly predicting the adjacent nodes to attend. Then, ANS-GT combines the weighted sampling heuristics to sample informative nodes.
>
> Directly predicting informative nodes for attention requires too much computational overhead and is hard to optimize. Comparatively, the pre-defined node sampling heuristics in our strategy help narrow the search space with prior knowledge. Moreover, the sampling strategy in ANS-GT can generalize to all nodes in the graph efficiently. Experiment results show that our strategy is effective and efficient.
>
> **Comment4:** The performance relies on the external graph coarsening method which is not updated during ANS-GT training.
>
> **Response4:** In the hierarchical attention scheme, we use graph coarsening algorithms to pre-process the input graph and generate a coarse graph for global attention. In ANS-GT, we do not update the coarse graph for training efficiency. In our future work, we may consider updating the coarse graph based on trained node embeddings for better performance.
>
> **Comment5:** It would be nice to see evaluations on graph tasks other than node classification.
>
> **Response5:** Previous studies have shown that Graph Transformer architectures such as Graphormer [1] have achieved state-of-the-art performance on graph classification. In this paper, we want to extend Graph Transformer to large graphs and focus on node classification tasks. Due to the page limits, we dedicate our efforts to provide a thorough evaluation of ANS-GT on node classifications and leave other graph tasks for future work.
>
> [1] Chengxuan Ying, Tianle Cai, Shengjie Luo, Shuxin Zheng, Guolin Ke, Di He, Yanming Shen, and Tie-Yan Liu. Do transformers really perform bad for graph representation? NeurIPS, 2021

---

> > ### Comment · Reviewer_vEEU · 2022-08-09
> > **I maintain my original rating.**
> >
> > Thanks the authors for your detailed reply. Most of my concerns have been addressed and I maintain my original rating.

---

### Author Response · Authors · 2022-08-07
**Response to Reviewers**

Dear reviewers, thanks again for your valuable and constructive suggestions! We have revised our paper following your suggestions and added more experiments to make our contributions clearer. Do you have further comments on our paper and responses?
If our responses have addressed your concerns, could you please kindly consider increasing the overall score?

---

### Author Response · Authors · 2022-08-09
**Thanks for the valuable suggestions, the revised paper has been uploaded.**

Dear reviewers, thanks for your appreciation and valuable suggestions. We are happy to know that most of your concerns have been addressed. We have included the answers and new results in our newly revised paper. We will keep polishing our paper to make the final version more comprehensive and solid.

---

### Meta-Review · Area_Chair_PUgU · 2022-08-28

**Recommendation:** Accept
**Confidence:** Less certain

**Metareview:**

The paper proposes to account for the graph properties when performing node sampling. The proposed method learns the effective sampling by adjusting the weights of a set of sampling schemes, based on an adversary bandit formulation. Besides, a hierarchical attention scheme for Graph Transformer is presented with graph coarsening algorithms. Overall the algorithm is sound and effective, as shown empirically on a range of benchmarks. The paper is well-written. The reviewers asked for comparison with other methods on more datasets, which the authors have partially resolved in their responses.

**Award:**

No

---

### Decision · Program_Chairs · 2022-09-14

Accept